# Artificial Intelligence-Based Software with CE Mark for Chest X-ray Interpretation: Opportunities and Challenges

**DOI:** 10.3390/diagnostics13122020

**Published:** 2023-06-10

**Authors:** Salvatore Claudio Fanni, Alessandro Marcucci, Federica Volpi, Salvatore Valentino, Emanuele Neri, Chiara Romei

**Affiliations:** 1Department of Translational Research, Academic Radiology, University of Pisa, 56126 Pisa, Italy; s.fanni1@studenti.unipi.it (S.C.F.); a.marcucci2@studenti.unipi.it (A.M.);; 2EBIT sr.l. Esaote Group, Via di Caciolle, 50127 Florence, Italy; 3Department of Diagnostic Imaging, 2nd Radiology Unit, Pisa University-Hospital, Via Paradisa 2, 56124 Pisa, Italy

**Keywords:** artificial intelligence, chest c-ray, CE-mark, deep learning, pulmonary tuberculosis, lung nodules

## Abstract

Chest X-ray (CXR) is the most important technique for performing chest imaging, despite its well-known limitations in terms of scope and sensitivity. These intrinsic limitations of CXR have prompted the development of several artificial intelligence (AI)-based software packages dedicated to CXR interpretation. The online database “AI for radiology” was queried to identify CE-marked AI-based software available for CXR interpretation. The returned studies were divided according to the targeted disease. AI-powered computer-aided detection software is already widely adopted in screening and triage for pulmonary tuberculosis, especially in countries with few resources and suffering from high a burden of this disease. AI-based software has also been demonstrated to be valuable for the detection of lung nodules detection, automated flagging of positive cases, and post-processing through the development of digital bone suppression software able to produce digital bone suppressed images. Finally, the majority of available CE-marked software packages for CXR are designed to recognize several findings, with potential differences in sensitivity and specificity for each of the recognized findings.

## 1. Introduction

Since the introduction of digital radiography in 1980, chest X-ray (CXR) has remained the most important imaging technique for chest imaging [1]. CXR is a cheap and easily accessible imaging technique with broad indications, characterized by short scan time and a lower dose compared to other imaging techniques, such as computed tomography, for example [2]. Nevertheless, CXR is limited in scope and sensitivity, and its interpretation is challenging due to the overlay of several different tissues belonging to a complex three-dimensional volume within a two-dimensional image [3].

The difference in density between pathological and healthy lung parenchyma may be subtle and challenging to detect, also because about 40% of lung parenchyma is covered by the ribs and mediastinum [4]. The experience of the radiologist and the cognitive process behind the interpretation of a CXR directly affect the clinical utility and effectiveness of CXR in patient management. To complicate matters even further, the rising demand for imaging exams, particularly in the emergency setting, may contribute to increasing numbers of errors [5].

The intrinsic limitations of CXR and the increasing number of examinations has prompted the proliferation of Artificial Intelligence (AI)-based software packages dedicated to CXR interpretation over the past decade [6].

AI-based image analysis, especially through deep learning, has become popular in recent years for classifying and segmenting the wealth of unlabeled data usually available in medical institutions in order to reduce, for example, the subjectivity and the time burden introduced by a manual labeling [7].

Still, to effectively tackle the above-mentioned issue, it is necessary to make these new technologies part of the daily routine. However, the gap between scientific research and clinical practice can be very significant. As a first essential step to achieving the clinical translation of all these research efforts, AI-based software needs to be certified for medical use. The providers affirm by means of CE marks the good conformity of the software with European health, safety, and environmental protection standards [8].

The primary aim of this review is to summarize the state-of-the-art of CE-marked AI-based software for CXR interpretation, focusing on their role, but also on the open issues of these applications and all the features of these software the reader must be aware of.

## 2. Materials and Methods

The online database “AI for radiology”, a non-commercial project tracking all CE-certified AI products, was queried to identify CE-marked AI-based software available for CXR interpretation [9]. Filters were applied to include only software developed for the chest, as subspecialty, and for X-ray, as imaging modality. No filters were applied regarding the level of CE and FDA (Food and Drug Administration) class. We ran the product and company name of each of the AI-based software packages through the PubMed database. Original research published in English was included in this review. The last search was run on 13 September 2022.

After collecting the studies, we used a narrative approach to analyze the state of the art of AI-based interpretation of different disease on CXR.

## 3. Results

We found 26 different CE-marked AI-based software packages.

The characteristics of the included software are summarized in Table 1. The studies were divided into three categories according to the targeted disease: AI-based software for tuberculosis-related abnormalities; AI-based software for nodule and lung cancer detection; and AI-based software for multiclass findings and other clinical scenarios. However, several software packages were not developed for a single disease or finding, and a significant overlap among categories was found.

### 3.1. AI-Based Software for Tuberculosis-Related Abnormalities

Pulmonary tuberculosis (TB) diagnosis is based on clinical features, sputum/blood test examinations positive for tubercle bacilli, and CXR presenting confluent dense shadows or signs of consolidation [10]. However, CXR as a screening and triage tool presents some relevant drawbacks, such as high inter- and intra-reader variability, moderate specificity, and limited radiology availability, especially in regions of the world with higher burdens of pulmonary TB [11].

Artificial intelligence (AI)-powered computer-aided detection (CAD) is already a fundamental tool in screening and triaging pulmonary TB, especially in countries suffering from a high burden of this disease. In these countries, CAD software packages are already widely adopted in mobile unit machines, which can operate without access to an electrical power grid [12]. Indeed, in 2021, the WHO recommended the use of CAD software for TB screening and triage, and the minimum required values of sensitivity and specificity were defined as 0.90 and 0.70, respectively [13,14].

However, the definition of software diagnostic accuracy is not that straightforward, as the performance may vary significantly according to several technical and clinical variables, see Figure 1. Furthermore, it is complicated to perform a direct comparison between the available software packages, or even between different versions (vrs) of the same software package. Only the newer vrs of the software CAD4TB (vrs 6 and vrs 7) and qXR (vrs 2 and vrs 3) utilize deep learning (DL) techniques and reach the required minimum values of sensitivity and specificity fixed by the WHO [13,14].

CAD software also differs in image processing and output. CAD4TB produces an abnormality heatmap indicating suspicious regions and provides a likelihood score ranging from 0 (no TB) to 100 (TB positive). Conversely, qXR compares the computed abnormality score with a prespecified threshold and provides a binary classification (positive or negative for pulmonary TB) [12,14]. This threshold must be tailored to achieve the optimal sensitivity and specificity according to the characteristics of the population studied, as suggested by Fehr et al., who achieved the best possible sensitivity for detecting pulmonary TB (84.8%), with a decisional threshold of 25 [15]. The proper tailoring of this threshold requires the consideration of clinical variables such as the TB burden in the studied population, diabetes, and HIV status [14]. 

According to the TB burden, we can differentiate high-burden and low-resource regions and low-burden high-resource regions. In high-burden low-resource regions (Pakistan, Nigeria, Brazil, etc.), the TB prevalence is almost 0.5%, but may reach as high as 2% when considering patients living with diabetes mellitus [16]. Mass screening surveys have been conducted in Karachi, Pakistan and Nigeria [17,18]. Due to the shortage of experienced CXR readers, in these regions, CAD could be used as the sole screening reader and may outperform the diagnostic accuracy of human reading [15]. In a resource-constrained setting, Philipsen et al. prospectively compared sputum testing alone and with the integration of CAD software for the selection of patients for sputum testing in terms of accuracy and cost. The results showed that the use of CAD software as a triage tool significantly reduced the cost, both per screened subject and per notified TB case, with only a few TB cases being missed [12,19]. Conversely, in low-burden high-resource countries, CAD software could be used to support triage or screening programs. Melendez et al. implemented CAD software in a screening program in London (UK) for a high-risk population including homeless people, prisoners and the drug and alcohol addicted [20].

Qin et al. compared the diagnostic accuracy of three DL algorithms (CAD4TB, Lunit and qXR), and concluded that the area under the curve (AUC) values for the three software packages were similar (0.92, 0.94, and 0.94, respectively), but most importantly, they pointed out the absence of a universal cut-off and the importance of identifying the proper threshold according to the TB prevalence in the population to achieve this level of performance [21].

However, even within the same region, there are different population subsets requiring specific adjustments. For instance, many studies have been conducted on prisoners [22,23]. The most significative study with respect to yield, efficiency and costs of mass screening algorithms for TB in prison was conducted in Brazil [24]. As a result, the integration of CAD4TB in this screening did not significantly change the cost per case diagnosed compared to other screening strategies without AI, and missed a higher number of cases. The most important drawback in this population subset was the high prevalence of HIV positivity and coinfection in the screened population [23,25].

Indeed, the detection of TB on CXR in HIV-positive subjects or patients with nontuberculous mycobacteria coinfection is quite challenging [19,26]. A strategy for avoiding this bias could be the stratification of the analysis according to the HIV status [19].

Two additional drawbacks to consider in the included studies are the significant heterogeneity regarding the gold standard and the methodology of comparison between algorithms and humans reader performance.

The majority of the included studies lack sputum culture testing for TB as a gold standard, usually because this test is expensive and requires several weeks to obtain a conclusive result, with rare exceptions [14,15,21,27]. As an alternative to sputum culture, the Xpert sputum test is usually adopted as a reference standard [12,26,28].

In regions where radiologists are not widely available for CXR reading, some studies have compared CAD software performance with the performance of clinical officers [29]. The difference between a clinical officer and an expert reader, such as a radiologist, was highlighted by Breuninger et al., who conducted a study with an older version of CAD4TB that outperformed only the clinical officers and not the experienced readers [26]. Moreover, only newer versions of CAD4TB and qXR achieved performance that was comparable to that of human radiologists in terms of triage ability [13,30]. A similar heterogeneity was reported by Qin et al., who assessed five AI algorithms during a screening campaign and also reviewed the literature related to CAD software for TB.

### 3.2. AI-Based Software for Nodule and Lung Cancer Detection

CXR remains, to date, the first line method for the diagnosis of lung nodules. Lung cancer benefits, in terms of survival, from early detection on CXR, but it has also been demonstrated that CXR fails to identify lung cancer in 18% of patients and that about 90% of mistakes in lung cancer diagnosis occur on CXR [31,32]. 

AI could assist radiologists in lung nodule diagnosis in several ways, such as detection, flagging of positive cases, and digital bone suppression.

A DL-based automatic detection algorithm has already been shown to be able to outperform the performance of physicians. However, in clinical practice, AI should be considered an aid to radiologists, and not an alternative [33].

Indeed, many papers have demonstrated that radiologists’ performance is enhanced when using AI as a second reader [34,35]. This benefit has been proven for young radiologists as well as for senior radiologists, thus being independent of reader experience [36]. However, as seen in the previous section on TB, the nature of this benefit may be affected by several technical and clinical–radiological variables, such as the nodule characteristics or the adopted ground truth.

AI-based detection performance in terms of accuracy and robustness seems to depend on the size and conspicuousness of the nodule solid portion. Indeed, inconspicuous nodules may cause fluctuations in the model output, leading to misclassification of findings [37].

Additionally, the algorithm performance may vary significantly according to the different adopted methods of ground truth. In a recent paper by Kim et al., a DL algorithm showed an AUC of 0.771 and 0.839 when using CT and radiologist readings as ground truth, respectively [38].

Liang et al. used the QUIBIM CXR Classifier to detect pulmonary nodules/masses. The four different algorithms of the software were analyzed and validated, and each one of them was characterized by different pros, such as higher sensitivity for the nodule probability algorithm or higher specificity for the heat map algorithm. Thus, to obtain the maximum benefits and minimize both false positives and false negatives, the radiologist may combine different algorithm strategies [39]. The increase in sensitivity, specificity and diagnostic confidence could be even higher for trained radiographers, which is particularly helpful in countries facing both a shortage of radiologists and high medical imaging demand [40].

Another application of AI exploiting the synergy between AI and radiologists in lung cancer diagnosis on CXR is the automated flagging of positive cases. For this purpose, Tam et al. proposed a triage workflow and achieved a significant reduction in missed cancer of 60% by using AI to flag positive case [41].

Moving to post-processing, to overcome the obscuration of lung cancer by overlying bone structures, digital bone suppression software has been developed, which is able to produce bone-suppressed images (BSIs). Digital bone suppression does not require special dual-energy hardware or additional dosage, and it is not affected by motion artifacts [42].

In 2013, Schalekamp et al. showed that BSIs improved radiologists’ detection performance for lung nodules compared to CXR alone (AUC = 0.883 vs. 0.855), particularly for nodules with moderate and subtle conspicuity [43].

However, BSIs have been shown to reduce specificity and lead to an increased overcall for focal lesions, which may result in unnecessary follow-up or further expansive investigations in normal patients [44,45]. It can be hypothesized that the lack of experience of the reader with BSIs may have caused this overcall, and it is likely that more practice will partially overcome this issue. 

Another possible solution is to combine both detection and post-processing strategies by applying CAD software after digital bone suppression. As reported by Schalekamp et al., CAD improved radiologists’ performance and detected 127 of 239 (53%) of the nodules that were initially missed by the radiologists. However, only 57 (45%) of these detections were accepted by the observer. According to the authors, the limited ability of the readers to reliably differentiate true-positive from false-positive CAD marks is currently one of the most important limits to CAD’s beneficial effect [46]. To optimize CAD’s beneficial effect and address this issue, two alternative methods already implemented with success in mammograms have been proposed for CXR. First, in the interactive mode, a CAD mark is shown together with a score of suspicions only if the radiologist clicks on the mark’s location. Secondly, the observer evaluates its score without viewing the CAD analysis, then a mathematical combination of the reader and the CAD score is computed. No improvement was seen with the interactive mode, while the mathematical combination significantly improved detection performance.

Thus, CAD provides information that might be even more useful when independently combined with a radiologist’s evaluation, without interfering with the reading process itself [47]. However, this approach could raise other legal and ethical problems, as the radiologist would have no way of checking the software output.

### 3.3. AI-Based Software for Multiclass Findings and other Clinical Scenarios

However highly accurate, the clinical benefit of a software package limited to a single or a small number of findings is at least questionable. The use of different software packages for different findings is unrealistic in clinical practice, and software covering the full range of CXR findings could be more effectively integrated into clinical practice.

Indeed, the majority of available CE-marked software packages for CXR fulfill this need, and are designed to recognize multiple findings. Multiclass finding detection must address differences in size and extension, ranging from subtle nodular opacities to extensive pleural effusion or pneumothorax [48]. Thus, the software may exhibit different performances for each finding, and the radiologist must be aware of this.

One clinical scenario necessarily requiring multiclass detection is the post-traumatic CXR performed in the emergency department, where findings vary from subtle ones to time-critical pathology [49]. In this setting, AI may aid in timely identification and worklist prioritization. Gipson et al. externally validated the software Annalise.ai for the detection of traumatic injuries on supine CXR, using CT as ground truth. In this study, the software outperformed radiologists for detection of pneumothorax and rib fractures, two of the most common pathologies in thoracic trauma, but was found to be inferior to radiologists for clavicle, humerus and scapula fractures [50]. However, as already pointed out in the previous paragraphs, comparing the performance of radiologists and AI is not enough to objectively investigate the beneficial effect provided by AI on clinical practice. An initial step was taken by Hwuang et al., who did not limit themselves to comparing the standalone detection performance of Lunit software and radiology residents in the emergency setting, but also assessed the effect of AI on residents’ performance. The results showed that when radiology residents used Lunit, the sensitivity improved from 65.6% to 73.4%, but, not surprisingly, a decrease in specificity was observed (98.1% vs. 94.3%) [51].

To evaluate the real-world usefulness of AI as a diagnostic assistance device for radiologists, Jones et al. set up a prospective multicenter study involving a group of radiologists using Annalise.ai in their daily reporting workflow [52]. Out of 2927 patients, 92 (3.1%) had significant reported changes, 43 (1.4%) had changed patient management, and 29 (1.0%) had further imaging recommendations. Moreover, radiologist attitudes towards the software were questioned, and almost all of them (9 out of 10) felt an improvement in their accuracy and a more positive attitude at the end of the study. 

Another relevant clinical scenario to address is pediatric radiology. As CT cannot be freely performed in children, CXR plays an even more critical role in pediatric radiology.

Shin et al. evaluated whether Lunit, an AI-based detection software developed and approved for adult CXR, could be used for pediatric CXR as well. The software assessed the presence of nodules, consolidation, fibrosis, atelectasis, cardiomegaly, pleural effusion, pneumothorax and pneumoperitoneum. Diagnostic accuracies up to 96.9% were found when cardiomegaly and children 2 years old or younger were excluded [53]. To detect cardiomegaly, other strategies may be explored. Bercean et al. trained two U-Net segmentation algorithms to contour the heart and the lungs and automatically compute the cardiothoracic ratio. This kind of approach reduced the reading time from 22.54 s to 5.14 s, and the F1 score for cardiomegaly detection was 0.85. However, AI may not necessarily reduce reading time, as showed by Kim et al., who reported a slight but significant increase in reading time from 14 s to 19 s when using AI [54].

Finally, COVID-19 pneumonia is a clinical scenario deserving of special mention, considering the ways in which it boosted radiology research, and particularly the AI field, and despite the SARS-CoV-2 pandemic is relatively recent, two different CE-certified deep-learning-based software packages are already available, qXR and CAD4COVID [55,56]. Mushtaq et al. investigated the prognostic utility of qXR on initial CXR. In this study, the software output was personalized to only report a score reflecting the percentage of pixels involved by opacity or consolidation, and a score ≥30 was shown to be an independent predictor for both mortality and severity of COVID-19 pneumonia (*p* < 0.001) [57]. The second software package is CAD4COVID, redesignated from CAD4TB vrs 6. CAD4COVID was efficiently integrated by Tovar et al. into a mobile TBC screening unit in Lima, Peru. Integrated TB and COVID-19 screening and testing services may ensure that TB case detection is maintained and not de-prioritized by health systems. The software produced a continuous abnormality score from 0 to 100, and all patients with a score >50 were tested for SARS-CoV-2 antibodies and by polymerase chain reaction (PCR) test, leading to a diagnosis of SARS-CoV-2 antibodies per three people screened and PCR-confirmed SARS-CoV-2 infection per eight people screened [58].

The radiologists’ interpretation of CXR, regardless of the clinical scenario, may be hindered by bone overlay. Thus, as well as for lung nodules, digital bone suppression may be helpful. BSIs have been demonstrated to be useful for improving radiologists’ detection performance for invasive pulmonary aspergillosis, single or multiple focal opacities, or signs of cardiac congestion [44,59]. These studies demonstrated that, without any significant negative effect on the interpretation of diffuse lung disease, the detection of focal chest abnormalities in CXRs, beyond just lung nodules, can be improved when their evaluation is aided by digital bone suppression.

### 3.4. Opportunities and Challenges

As highlighted in the previous paragraphs, the number of publications in the field of AI and of the available commercial applications for CXR interpretation is steadily increasing. Radiologists will definitively have to deal with these solutions, and therefore must be aware of both the opportunities they offer as well as their challenges.

Artificial intelligence could be the answer to the problem of an ever-increasing population faced by healthcare systems and the rising number of examinations referred to the radiology departments [60].

These software packages may help in finding and detecting various types of CXR abnormalities, saving work for the radiologists and time that can be dedicated to harder issues. This is possible today thanks to the availability of software covering the full range of CXR findings, from fractures to pleural effusions. In those regions in which expert radiologists may be lacking, such as developing countries, these software packages may really be game changers.

The integration of AI as a second reader is a topic of extreme interest and an emergent subject of study. AI has been demonstrated able to significantly increase radiologists’ sensitivity; however, it is also proven to decrease specificity, especially in unexperienced radiologists. This dual effect makes necessary to emphasize that the responsibility of final diagnosis still lies with the radiologists, not AI systems, [61] and that radiologists must be aware of when to trust the AI output.

An additional point of complexity concerns the previously mentioned AI-based software for multiclass findings. The software may present different levels of sensitivity and specificity for each of the recognized findings, and the radiologist must be aware of this in order to trust the different outputs with more or less confidence.

Some issues still undermine the building of a trust relationship between radiologists and AI, significantly slowing down its integration into clinical practice. One of these issues is the lack of explainability of the software and the lack of transparency of the available literature, particularly regarding algorithm and dataset characteristics and limitations [62].

However, promising progress has been achieved in recent years, such as the increasing number of papers reporting objective comparisons between different AI software packages performed with independent datasets.

Another relevant issue is the lack of large-scale multicentric datasets, which are pivotal for training the algorithms, and this is particularly true for pediatric CXR. Pediatric imaging requires expertise and also dedicated or adaptable equipment [63]. Similarly, AI-based software requires specific training to achieve acceptable performance for pediatric CXR interpretation, and the application of AI in this promising field is currently limited.

## 4. Conclusions

CXR is still the most important imaging technique for chest imaging, as evidenced by the number of published studies and software packages that have been developed for this technique. AI-based software packages that have obtained CE markings have been created for diverse clinical scenarios, including TB screening and nodule detection. AI was demonstrated to be able to improve the diagnostic accuracy of CXR in TB screening, with performance varying according to the software characteristics and the targeted population. In this setting, to be aware of this dependence is critical to properly tailoring the software according to the clinical context. AI-based software also affirmed its role as a second reader when performing nodule detection by CXR. Nodule features and combination BSIs seem to have more impact on diagnostic performance than the level of experience of the user. Finally, AI-based software for the detection of multiclass findings was developed to comprehensively address the complexity and wide range of potential findings.

In conclusion, the primary objective of this software is not to outperform the diagnostic performance of radiologists, but rather to enhance it. In order to achieve this objective, the radiologist must be aware not only of the benefits but, more significantly, of the limitations of this software.

## Figures and Tables

**Figure 1 diagnostics-13-02020-f001:**
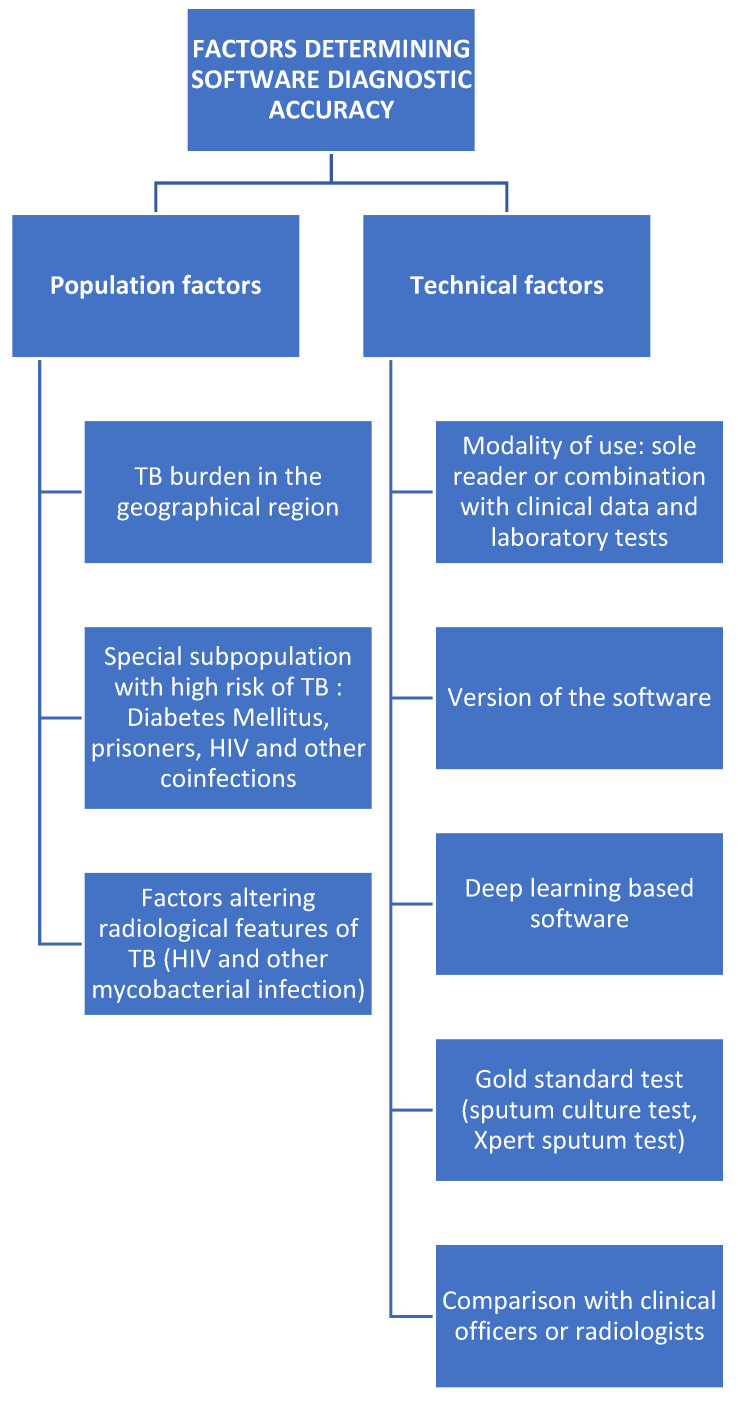
Factors determining software diagnostic accuracy.

**Table 1 diagnostics-13-02020-t001:** Characteristics of the included Software (information provided from the online database “AI for radiology”).

Software Name	Company	Disease Targeted	Population	Processing Time
ClearRead XRay	Riverain Technologies	Lung cancer	Adult, not pediatric	1–10 min
ChestLink	Oxipit	75 different pathologies	Older than 18 years old	10–60 s
CAD4TB	Thirona	Tuberculosis	All chest X-rays	3–10 s
AI-Rad Companion Chest X-ray	Siemens Healthineers	Pulmonary lesions, pleural effusion, pneumothorax, consolidation, atelectasis	All upright chest X-rays	1–10 min
SenseCare-Chest DR Pro	SenseTime	Pneumonia, tuberculosis, pneumothorax, pleural effusion, cardiomegaly, rib fractures	All chest X-rays	3–10 s
Annalise Enterprise CXR	annalise.ai	124 findings present in the emergent, urgent, and non-urgent care settings	Patients over 16 years of age	3–10 s
Chest | MSK AI	Arterys	Fracture, dislocation, elbow joint effusion, pleural effusion, pulmonary nodule, pulmonary opacity, pneumothorax	Emergency department population	Not available
Chest X-ray Classifier	Quibim	14 different findings	All chest X-rays	10–60 s
qXR	Qure.ai	Tuberculosis, COVID-19, signs seen in Lung Parenchyma, Pleura, Mediastinum, Cardiac and bones	All chest X-rays	10–60 s
TIRESYA	Digitec	Unspecified	Unspecified	Not available
Milvue Suite	Milvue	Bone fracture, pleural effusion, lung opacity, elbow joint effusion, lung nodule, pneumothorax, dislocation	All chest X-rays	10–60 s
CheXVision	XVision	17 pathologies	All chest X-rays of patients above 16 years old	3–10 s
CAD4COVID-XRay	Thirona	COVID-19	All chest X-rays	3–10 s
Chest Solution	Nanox.AI	Pneumothorax, pleural effusion	Unspecified	Not available
VUNO Med^®^-Chest X-ray™	VUNO	Nodule/Mass, Consolidation, Interstitial Opacity, Pneumothorax, Pleural Effusion	All chest X-rays	<3 s
InferRead DR Tuberculosis	Infervision	Tuberculosis	Unspecified	Not available
InferRead DR Chest	Infervision	Lung cancer, pneumothorax, fracture, tuberculosis, lung infection, aortic calcification, cord imaging, heart shadow enlargement, pleural effusion.	Any	Not available
X1	Visionairy Health	12 different findings	Unspecified	Not available
Auto Lung Nodule Detection	Samsung Electronics	Lung cancer	Unspecified	<3 s
Pneumothorax (Ptx)	Aidoc	Pneumothorax	Radiograph	1–10 min
Red Dot	Behold.ai	Pneumothorax	Unspecified	Not available
JLD-02K	JLK Inc.	Lung cancer	All Chest X-rays	3–10 s
Lunit INSIGHT CXR	Lunit	11 different findings	Patients aged 14 years or older	3–10 s
Critical Care Suite	GE Healthcare	Pneumothorax	Adult-size patients. Suspected of pneumothorax or intubated.	<3 s
ChestView	GLEAMER	Pneumothorax, pleural effusion, alveolar syndrome, nodule, mediastinal mass	All Chest X-rays	10–60 s
ChestEye Quality	Oxipit	75 different pathologies	Patients over 18 years old	3–10 s

## Data Availability

Not applicable.

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
