# Peer review of "Artificial Intelligence-Based Software with CE Mark for Chest X-ray Interpretation: Opportunities and Challenges"

_diagnostics, 2023, doi:10.3390/diagnostics13122020_

Round 1

Reviewer 1 Report

Thank you for the opportunity to review this manuscript.This article reviews CE-marked AI-based software available for CXR interpretation. I am very interested in this. However, there are still some areas that need to be revised and supplemented.

1.The opportunities and challenges in the title of the article should be addressed in the special chapter.

2.At the beginning of the article, it was introduced that the technology of CXR and the challenges for doctors to read CXR were the conclusions before the research on CE-marked AI-based software available for CXR interpretation. It is recommended to add deeper understanding of the opportunities and challenges after researching 60 articles.

3.Table 1 in the article is well done, presenting interesting information about Software name, Company, Disease targeted , and Population. It is recommended to present the article content in more graphical and statistical ways.

4.The section 3.1, 3.2, and 3.3 focus on describing the role of the latest software technology in a narrative way, which is somewhat monotonous. It is recommended to summarize and refine.

5.Lines 76 to 90 are all describing diseases, which takes up too much space in the main part of the article. It is recommended to concise and brief the introduction.

6.The clinical effects and current functional limitations of these CE-marked AI-based software available for CXR interpretation, as well as the evaluation of their use by doctors, are very interesting. Could the author provide additional information.

7.There are Companys and Population in Table 1, but there is no analysis of these two types of data in the text. It is recommended to supplement them.

8.Although the topic of the special issue is chest X-ray, there are many ways to detect pulmonary nodules using CT. It is recommended that the article fully explain the role of software in this regard.

9.The articles research on CXR combined with CT for diagnostic software can make the article more convincing. It is recommended that the author supplement more CT content.

10.The conclusion of the article is that doctors should be aware of the limitations of software and that the software is mainly intended to assist doctors. The opportunities and challenges presented by the software described in the article can also be for researchers. It is recommended that the author provide additional explanations from different perspectives.

Author Response

Response to comment 1: We added a chapter regarding opportunities and challenges. 

Response to comment 2: For a deeper understanding on the opportunities and challenges we added a dedicated paragraph: 3.4 “opportunities and challenges”.

Response to comment 3: Thank you for the suggestion. We added Figure 1 in chapter 3.1 “AI-based software for tuberculosis-related abnormalities” for a better understanding.

Response to comment 4: We hope that the addition of figure 1 will facilitate the reading of chapter 3.1.

Response to comment 5: We rephrased the introduction in a more pertinent and concise way.

Response to comment 6: We also believe that more information on clinical effects and current functional limitations of these software are very interesting, but we found little literature and that’s why we added this topic on the chapter “Opportunities and challenges”.

Response to comment 7: We added a reference to table 1 for highlight population data in the chapter 3.4.  

Response to comment 8: Even if it is an interesting comparison, detection of pulmonary nodules using CT goes beyond the purpose of the review.

Response to comment 9: Even if the combination CXR and CT for diagnostic software is an interesting topic it goes beyond the purpose of the review.

Response to comment 10: The articles we explored are mainly focused on clinical perspectives. Researchers’ point of view goes beyond the goal of this review.

Reviewer 2 Report

This paper titled “Artificial Intelligence-based software with CE mark for chest X- rayinterpretation: and challenges” presents an overview on the CE-Marked AI software for X-ray image, this topic is interesting, However, I’d like to raise some issues,

1.      There has been many works focusing on classification/ segmentation based on deep learning, such as, Context-aware Network Fusing Transformer and V-Net for Semi-supervised Segmentation of 3D Left Atrium, https://doi.org/ 10.1016/j.eswa.2022.119105, and others. The authors can first give a brief introduction on AI based image interpretation, then transfer to X-ray chest image analysis.

2.     The authors just gave some overview on the CE-Marked AI softwares, and listed 26 ones. It is somewhat limited just focusing on software, if the authors broaden the content to X-ray Chest image analysis based on AI, not just the AI software, it is much better.

3.     Meanwhile, the title includes “opportunities and challenges”, however, the authors did not discuss the opportunities and challenges.

Author Response

Response to comment 1:  We revised the introduction based on your interesting suggestion

Response to comment 2:  We thank the reviewer for the kind suggestion and we agree that different models of AI image analysis would be interesting to describe, however these information are generally not reported for the software revised. 

Response to comment 3:  We added a chapter regarding opportunities and challenges.

Reviewer 3 Report

Authors have proposed a review paper based on the AI bases X-Ray interpretation. 

You have not highlighted any opportunities or any challenges.

The style of writing this review paper is not impressive

Even though many references have been cited but the clarity or any novelty could not be found,

Some references have been cited several time with difference perception. Ex. [13]n 3 times with three difference perspective.

[15] - 5 times. This happens many times.

Conclusion is not impressive.

Author Response

We added a chapter on opportunities and challenges, we hope it will help highlighting the key points of the review.

Reviewer 4 Report

I am really grateful for reviewing this manuscript. In my opinion, this manuscript can be published once some revisions are done successfully. This study reviewed 26 artificial intelligence-based software to examine and interpret chest x-ray data. I would argue that this is a rare achievement. However, it needs to be noted that different deep learning models would lead to different combinations of performance and execution time hence their information would be really helpful to choose the best software available for specific clinical needs. In this context, I would suggest the authors to include information on deep learning models, their performances and their execution times in Table 1. 

Author Response

We thank the reviewer for the kind suggestion, we added the processing time of the software in the table. We also agree that different deep learning models would lead to different combinations of performance, however this relevant information is generally not reported in the articles revised, this leading to our choice to exclude this information from the Table 1.

Round 2

Reviewer 1 Report

The main revision of the article involves adding opportunities and challenges. In response to comments, the authors clarify that the article mainly addresses clinical perspectives. However, as there is no feedback from physicians using the software, the discussion on clinical perspectives remains declarative. The article relies on descriptions of the software's functions and algorithms in the paper, but there is insufficient explanation from both a software and algorithmic perspective.

Reviewer 3 Report

Please summarize the survey with the similarity of the work together and make a analysis how the research progress in the area of study. That is group the works bases on the similarity and elaborately discuss the area in which the study progresses fast and attract the researchers.

Reviewer 4 Report

I am really grateful to review this manuscript. In my opinion, this manuscript can be published in current form.